# Development of label-free cell tracking for discrimination of the heterogeneous mesenchymal migration

**Sota Endo, Shotaro Yamamoto, Hiromi Miyoshi** [ID]*

Department of Mechanical Systems Engineering, Graduate School of Systems Design, Tokyo Metropolitan University, Hachioji, Tokyo, Japan

* hiromi-miyoshi@tmu.ac.jp

## Abstract

Image-based cell phenotyping is fundamental in both cell biology and medicine. As cells are dynamic systems, phenotyping based on static data is complemented by dynamic data extracted from time-dependent cell characteristics. We developed a label-free automatic tracking method for phase contrast images. We examined the possibility of using cell motility-based discrimination to identify different types of mesenchymal migration in invasive malignant cancer and non-cancer cells. These cells were cultured in plastic tissue culture vessels, using motility parameters from cell trajectories extracted with label-free tracking. Correlation analysis with these motility parameters identified characteristic parameters for cancer HT1080 fibrosarcoma and non-cancer 3T3-Swiss fibroblast cell lines. The parameter "sum of turn angles," combined with the "frequency of turns" at shallow angles and "migration speed," proved effective in highlighting the migration characteristics of these cells. It revealed differences in their mechanisms for generating effective propulsive forces. The requirements to characterize these differences included the spatio-temporal resolution of segmentation and tracking, capable of detecting polarity changes associated with cell morphological alterations and cell body displacement. With the segmentation and tracking method proposed here, a discrimination curve computed using quadratic discrimination analysis from the "sum of turn angles" and "frequency of turns below 30°" gave the best performance with a 94% sensitivity. Cell migration is a process related not only to cancer but also to tissue healing and growth. The proposed methodology is easy to use, enabling anyone without professional skills in image analysis, large training datasets, or special devices. It has the potential for application not only in cancer cell discrimination but also in a broad range of applications and basic research. Validating the expandability of this method to characterize cell migration, including the scheme of propulsive force generation, is an important consideration for future study.

## Introduction

Image-based cell phenotyping is a fundamental methodology in both cell biology and medicine. Conventionally, cell phenotypes are characterized based on static information, such as morphology and protein localization [1,2]. However, cells are dynamic systems that undergo

**Data availability statement:** All relevant data are within the manuscript and its Supporting Information files.

**Funding:** This work was supported by JSPS KAKENHI Grant Number JP24K01998, Tokyo Metropolitan Government Advanced Research Grant Number R2-2, and TMU strategic research fund for social engagement. The funders had no role in study design, data collection and analysis, decision to publish, or preparation of the manuscript.

**Competing interests:** The authors have declared that no competing interests exist.

a diverse range of processes, such as biochemical signaling (1–10 s), transcriptional change (minutes to hours), cell migration (minutes to hours), and the cell cycle (one day) [3]. Cell phenotyping based on static data can be complemented by dynamic data extracted from time-dependent cell characteristics [4].

Cell migration is a fundamental process in multicellular pattern formation, wound healing, and cancer metastasis [5–7], and thus will serve as a good indicator for characterizing the cellular state during these processes. A typical example is epithelial-to-mesenchymal transition (EMT), which is crucial in cancer progression and metastasis, as well as in normal development; changes in gene expression and post-translational regulation result in the repression of epithelial characteristics and the acquisition of mesenchymal traits, increasing motility capacity. The cell motility phenotype is not limited to the binary classification of epithelial-mesenchymal [8]. Instead, it represents a finely tuned spectrum, as shown in research that successfully predicted the lineages of skeletal muscle stem cells [9] and hematopoietic cells [10] based on their motile behavior. Classifying cells based on subtle differences in their finely tuned migration machinery not only provides valuable applications in research and diagnostics but also offers important biomedical insights into the physiological and signaling states associated with cell migration machinery [11–13]. As this machinery involves chemical and mechanical processes that interact across multiple spatial and temporal scales, leading to heterogeneous behaviors on various levels [14], the spatiotemporal scales of data acquisition are vital for effectively processing and extracting discriminative characteristics.

Effective segmentation and tracking methods are limited by the spatiotemporal resolution and observation duration, which depend on the image acquisition method. While conventional manual tracking methods [15–17] perform well across a wide range of image acquisition methods, they generally require selecting only a subset of cells from each time-lapse image for analysis, which is assumed to represent the whole cell population. Such manual selection of cell subsets potentially introduces significant bias [18]. Therefore, high-throughput automatic tracking has the advantage of improving the effectiveness and reliability of cell motility analyses. Labeling cells with fluorescence is an option for cell segmentation and tracking, though the labeling procedure can sometimes influence cell behavior unless experimental conditions are carefully optimized [19–21]. Additionally, prolonged exposure to excitation light raises concerns about phototoxic side effects [22]. To address these concerns, phase contrast and differential interference contrast (DIC) microscopy are two widely used label-free techniques capable of producing high-contrast images of cells. Several review articles [23–25] have summarized and compared tools for automated segmentation and tracking of cells recorded using various imaging modalities, including phase contrast and DIC microscopy. Birefringent specimen containers, such as commonly used plastic tissue culture vessels, depolarize light and compromise DIC images, whereas phase contrast microscopy remains unaffected. However, phase contrast images often contain halo and shade-off artifacts. When using tracking tools for phase contrast images [26–30], it is important to consider the limitations of spatial resolution. The halo effect, particularly crucial in thick specimens, obscures detailed cell morphology. For applications focused solely on cell body displacement, the halo effect poses no issue and can even enhance contrast, aiding in the detection of specimens. In cases involving cells with a spreading shape on two-dimensional adhesive surfaces, such as conventional tissue culture polystyrene, the halo effect is minimal. This facilitates well-resolved cell contours at a subcellular scale, especially when combined with image analysis methods capable of detecting the low contrast associated with small phase shifts [29,30].

In this study, we aimed to investigate the potential to differentiate between the cancer cell line HT1080 fibrosarcoma and the non-cancer cell line 3T3-Swiss fibroblast. Both cell lines exhibit mesenchymal mode migration when migrating as single cells [5]; however, their

mesenchymal mode migrations are assumed to display heterogeneity due to differences in malignancy, such as the presence or absence of invasiveness. Mesenchymal morphology is characterized by an elongated or polygonal cell shape with features such as filopodia, lamellipodia, and/or ruffles (detached lamellipodia) [31]. In phase contrast images, the thinly elongated filopodia and thinly spreading lamellipodia often exhibit low contrast, attributed to the small phase shifts associated with these structures. We propose an easy-to-use, label-free, semi-automatic segmentation and tracking method for phase contrast images. This method was specifically applicable to phase contrast image recordings of mesenchymal mode migration in label-free cells every minute for 30 h, including those with low-contrast filopodia and lamellipodia. We further performed discriminant analysis using acquired motility parameters. We identified the parameters to characterize each cell type. Using the identified parameters, linear and quadratic discriminant analyses, which are fundamental classification methods, were conducted to distinguish the mesenchymal mode migrations of HT1080 fibrosarcoma cells from that of 3T3-Swiss fibroblasts for validation of this methodology. Additionally, based on the results of the discriminant analysis, we identified the requirements for segmentation and tracking methods necessary to effectively process and extract the characteristics of mesenchymal mode migration. This includes capturing detailed differences to distinctly differentiate between the malignant HT1080 fibrosarcoma cells and the non-malignant 3T3-Swiss fibroblasts.

## Materials and methods

### Cell culture and imaging

We used HT1080 fibrosarcoma cells (ATCC CCL-121) as a cancer cell model and 3T3-Swiss fibroblasts (JCRB 9019) as a non-malignant normal cell model for the validation of our tool. Cells were cultured in a humidity-controlled environment under 5% $CO_2$ at 37 °C, using Minimum Essential Medium α (Gibco) supplemented with 10% fetal bovine serum (HyClone), 100 units/mL penicillin, and 100 μg/mL streptomycin (FUJIFILM Wako Pure Chemical Corporation). Cells were seeded at a density of 2,000 cells per cm² in a polystyrene dish (35 mm, VIOLAMO) and allowed to attach for 24 h. After 24 h, live cell imaging was performed every 1 min for 30 h with phase contrast microscopy by using the following devices: a CMOS camera (ORCA-Flash 4.0, Hamamatsu Photonics) controlled by the cellSens software (Olympus), a 10 × 0.30 NA objective lens (UPLFLN10X2, Olympus), and a stage-top incubator (TOKAI HIT) on an inverted microscope (IX83, Olympus). For imaging, the cells were incubated at 37 °C in an atmosphere of 5% $CO_2$.

### Cell tracking

Image analyses were performed with MATLAB R2020a. As shown in Fig. 1, we tracked cell migration trajectories by detecting cell contours from the phase contrast images and calculating centroids. The flowchart of automated cell tracking is shown in Fig. 1A. First, we segmented each of the cells from the image containing multiple cells (1. cell segmentation); then, we calculated the centroids of all the cells in the image (2. calculation of cell centroids). Next, we drew a square region of interest (ROI) by cropping the image including a target cell with the cell centroid at the center of the image (3. drawing of the square ROI including a target cell in the center). The cell centroid and the major axis of the cell were calculated for the next time point in the ROI and were recorded (4. calculation of the centroid in the ROI). Then, we reshaped and moved the ROI to the next time point (5. moving the ROI to include the target cell with the centroid in the center). From the second time point, the ROI was set in reference to the centroid and major axis in the image at the previous time point to capture the targeted

(A)

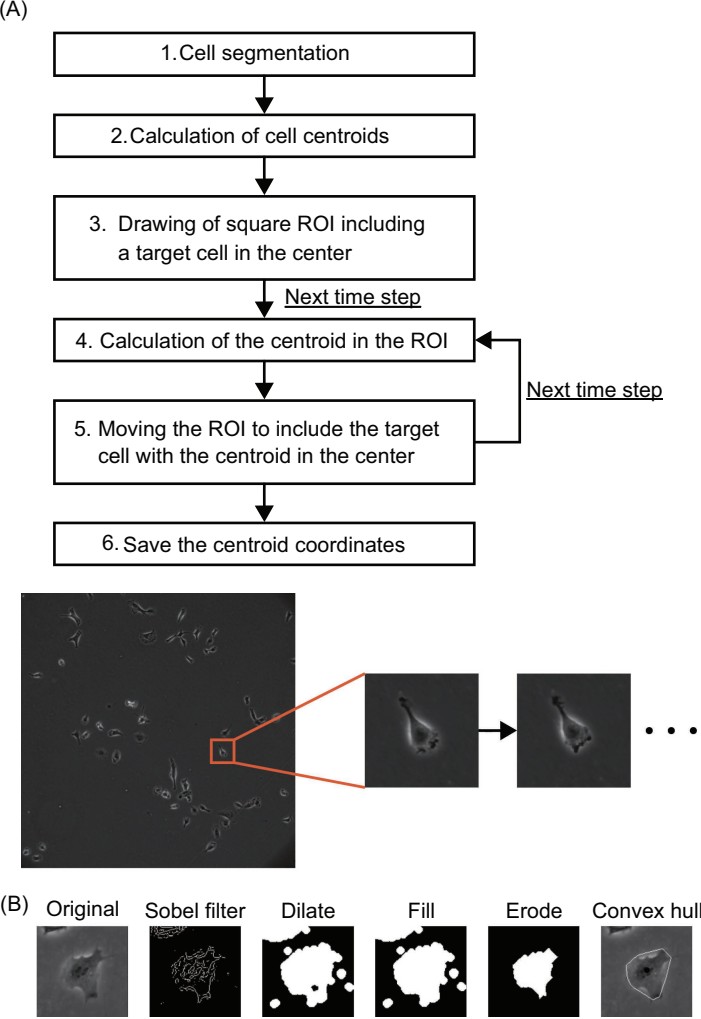

(B)

**Fig. 1. Method for cell segmentation and tracking. (A)** Flowchart of automated cell tracking applied to a time series of phase contrast images acquired with a $10\times$ objective lens. The method involved shifting the ROI while adjusting the shape by referring to the centroid and cell size over time, allowing for single cell tracking. **(B)** Cell segmentation and centroid calculation process from the phase contract image. The cell spreading area was detected using the Sobel filter, and the cell outline was extracted from the convex hull. Scale bar: 50 μm.

cell in the center of the ROI. The process was applied to each of the cells in the time series of the phase contrast image. The tracking was stopped when cell division or contact with other cells occurred. In such instances, either two centroids or no centroids were computed, which served as the criterion for stopping the tracking. For analysis, only the tracking data collected prior to the point of contact were used.

The image processing for cell segmentation and centroid calculation is shown in Fig. 1B. First, the cell-spreading area was detected using a Sobel filter, which is a first-order differential filter. The detected edges were then binarized using a threshold. Once the threshold was manually adjusted for an image, the same value was applied regardless of whether it was HT1080 fibrosarcoma cells or non-cancer 3T3-Swiss fibroblasts for the same image acquisition setup. Next, the binarized images were dilated, and holes were filled. The dilation was performed using an octagonal structuring element with a radius of 3 pixels (1.95 μm) from the origin to

the sides of the octagon, measured along the horizontal and vertical axes. In addition, erosion was performed twice using a diamond structuring element with 5 pixels (3.25 μm) from the origin of the structural element to the point of the rhombus. The cell centroid was calculated against the smoothed cell outline of the convex hull, which was the smallest convex polygon encompassing all the given points.

## Cell motility parameters

Cell motility parameters were calculated from the centroid data acquired. These parameters included migration speed, frequency of turns below thresholds, migration distance between turns below thresholds, sum of turn angles, time to stop, and total migration length (S1 Table). The migration speed was calculated from the displacement of the centroids. Turn angles, defined as shown in Fig. 2, were calculated from the centroids at three time points. In cases where there was no displacement in 1 min (i.e., the migration speed = 0), the turn angle was set to 0. The frequency of turns and migration distance between turns below each of the three turn angle thresholds (90°, 60°, and 30°) were determined. Instances where the migration speed = 0 were excluded from the count of the "frequency of turns" and the "migration distance between turns below the threshold." Quiescent time was defined as the duration when speed was lower than a defined threshold, and the threshold was the average speed. The sum of the turn angles was calculated from the turn angles. The total migration length was the sum of the displacements of the centroids. Data that could be tracked for more than 1 h were

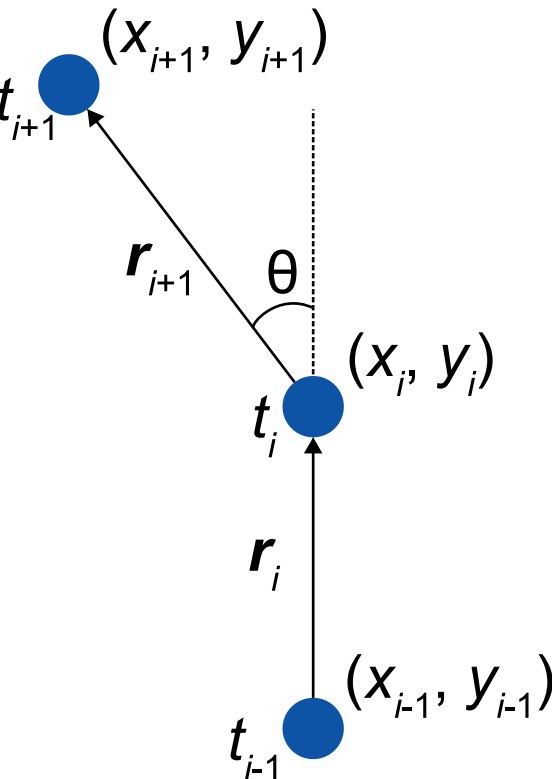

**Fig. 2. Definition of the turn angle.** The turn angle $\theta$ was calculated from the cell centroids ($x_{i-1}$, $y_{i-1}$), ($x_i$, $y_i$), and ($x_{i+1}$, $y_{i+1}$), respectively, at times $t_{i-1}$, $t_i$, and $t_{i+1}$ as the angle, $\theta$, between the displacement vectors, $r_i$ and $r_{i+1}$.

used for analysis. Parameters such as the frequency of turns below a threshold, the sum of turn angles, and the total migration length were influenced by the total tracking time. These parameters were normalized to values per 30 h by multiplying the hourly values by 30 for statistical and discrimination analysis.

## Correlation coefficient

Correlation coefficients were calculated using the following formula:

$$r = \frac{s_{xy}}{s_x s_y} \tag{1}$$

where $s_{xy}$ is covariance, and $s_x$, $s_y$ are respectively standard deviations of $x$ and $y$. The average values of the parameters were used to calculate the correlation coefficients for each cell.

## Discriminant analysis

We performed a discriminant analysis of the motility parameters. Linear discriminant analysis (LDA) and quadratic discriminant analysis (QDA) were used to discriminate and compare the discriminant models based on their sensitivity, specificity, and accuracy (Table 1) according to Eqs. (2)–(4):

$$Sensitivity = \frac{\text{True positive}}{\text{True positive} + \text{False negative}} \tag{2}$$

$$pecificity = \frac{\text{True negative}}{\text{True negative} + \text{False positive}} \tag{3}$$

$$Accuracy = \frac{\text{True positive} + \text{True negative}}{\text{True positive} + \text{False negative} + \text{True negative} + \text{False positive}} \tag{4}$$

Then, k-fold cross-validation with $k = 10$ was performed to evaluate the generalization performance of the discriminant model.

LDA is a method used to discriminate between the variance and average of the data. The variance-covariance matrices were calculated for the set of parameters $x_i = (a_i, \ b_i)^t$, obtained for each of the two cell types $(i = 1, \ 2)$. The variance-covariance matrices $S_i \ (i = 1, \ 2)$, were obtained using Eq. (5):

$$S_i = \begin{pmatrix} Var[a_i] & Cov[a_i, b_i] \\ Cov[a_i, b_i] & Var[b_i] \end{pmatrix}, \ i = 1, 2 \tag{5}$$

**Table 1. Confusion matrix.**

| | | Actual | |
|---|---|---|---|
| | | Positive | Negative |
| Predicted | Positive | True positive | False positive |
| | Negative | False negative | True negative |

where $Var[a_i]$ and $Var[b_i]$ represent the variance and $Cov[a_i,b_i]$ the covariance. In LDA, the variance-covariance matrices $S_i$ of the two cell types ($i = 1, 2$) were assumed to be equal, and $S$, which pooled the variance and covariance of the two groups, was obtained using Eq. (6):

$$S = \frac{(n_1-1)S_1 + (n_2-1)S_2}{(n_1-1) + (n_2-1)} \tag{6}$$

Furthermore, $n_1$ and $n_2$ are the sample sizes of HT1080 and 3T3-Swiss fibroblasts, respectively. The Mahalanobis square distance was calculated using Eq. (7):

$$d_i^2 = (x - \bar{x}_i)^t S^{-1} (x - \bar{x}_i), \; i = 1,2 \tag{7}$$

and when $d_1^2 - d_2^2 \geq 0$, the cells with parameter $x = (a,b)^t$ were regarded as positive.

In QDA, unlike LDA, the variance-covariance matrices of the populations of the two groups are assumed to be different, and Eq. (8) was used for discrimination.

$$d_1^2 - d_2^2 = x^t \left( S_1^{-1} - S_2^{-1} \right) x - 2x^t \left( S_1^{-1} \bar{x}_1 - S_2^{-1} \bar{x}_2 \right) + \left( \bar{x}_1^t S_1^{-1} \bar{x}_1 - \bar{x}_2^t S_2^{-1} \bar{x}_2 \right) \tag{8}$$

As in LDA, when $d_1^2 - d_2^2 \geq 0$, cells with parameter $x = (a,b)^t$ were classified as positive.

## Results

### Cell segmentation and tracking

To develop a label-free automatic tracking method and evaluate its potential for motility-based cell type discrimination, we applied this approach to HT1080 fibrosarcoma cells as a cancer cell model and 3T3-Swiss fibroblasts as a normal cell model.

The results of the cell segmentation are shown in Fig. 3A and 3B, and S1 and S2 Videos. Our method achieved good cell segmentation and tracking in both cancer (HT1080 fibrosarcoma cells) and non-cancer (3T3-Swiss fibroblasts) cells. The average equivalent circle diameters were 47.2 ± 9.7 μm (mean ± SD, n = 50) for HT1080 fibrosarcoma cells and 52.2 ± 7.9 μm (mean ± SD, n = 50) for 3T3-Swiss fibroblasts. The average spreading areas were 1865.9 ± 871.8 μm² (mean ± SD, n = 50) for HT1080 fibrosarcoma cells and 2251.8 ± 618.7 μm² (mean ± SD, n = 50) for 3T3-Swiss fibroblasts. The trajectories of the cell centroids obtained by automatic and manual tracking are shown in Fig. 3C and D. The trajectories are nearly the same, and the differences in the coordinates of the cell centroids were within 2–3 μm for both cell types (Table 2). Occasionally, as shown in Fig. 3D, the coordinates obtained by automatic tracking deviate from those obtained by manual tracking. This occurred mainly when a cell showed an elongated and droplet-like shape, probably because the centroids for such shaped cells tended to be misleading by visual evaluation in the manual tracking. With this process, the smoothing process did not correctly extract the outline of the cell and exhibited a constricted shape (S1A Fig.). Moreover, if cell–cell contacts occurred during cell tracking, this method recognized the cells as a single cell (S1B Fig.). In these cases, where either no centroid or more than one cell centroid was detected, cell tracking was stopped automatically at that point and the tracking data before contact were used for analysis. In the automatic tracking of the live cell imaging data for 30 h, the average tracking time for the data used in this analysis was 3.3 and 4.4 h for HT1080 (n = 50 cells) and 3T3-Swiss (n = 50 cells) cells, respectively, and the maximum time length was 13.6 and 20.6 h, respectively.

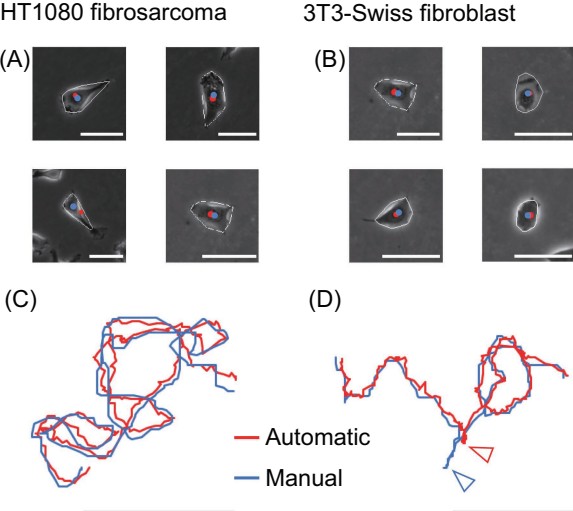

**Fig. 3. Label-free cell segmentation using automatic tracking.** Representative results for an HT1080 cell **(A, C)** and a 3T3-Swiss fibroblast **(B, D)**. **(A, B)** Typical results of automatically segmented cell outlines (white lines). Blue dots are cell centroids calculated from the automatically segmented cell outlines. Red dots are the center positions of the cells estimated with the naked eye. **(C, D)** Cell trajectories. The results with automatic tracking (red lines) are superimposed with those with manual tracking (blue lines). Scale bars: 50 μm.

**Table 2. Difference in the cell centroid coordinate.**

|  | $\left\|x_{auto} - x_{manual}\right\|$ [μm] | $\left\|y_{auto} - y_{manual}\right\|$ [μm] |
| --- | --- | --- |
| HT1080 | 2.31 ± 1.88 | 2.09 ± 1.68 |
| 3T3-Swiss | 3.00 ± 2.31 | 2.12 ± 1.63 |

$x_{auto}$, $y_{auto}$ : automatically acquired coordinates of the cell centroid. $x_{manual}$, $y_{manual}$ : cell center manually determined with the naked eye.

## Quantitative characterization of motilities for HT1080 fibrosarcoma cells and 3T3-Swiss fibroblasts

Five typical trajectories of HT1080 cells and 3T3-Swiss fibroblasts, acquired by the automatic tracking are shown in Fig. 4. HT1080 cells were more motile and directional than 3T3-Swiss fibroblasts.

Fig. 5A and B, respectively, shows typical time courses of migration speed in HT1080 fibrosarcoma cells and 3T3-Swiss fibroblasts. HT1080 migration displayed two-speed modes, consisting of transient high speed and slow speed for the rest of the time, whereas 3T3-Swiss migration speed was temporally homogeneous (Fig. 5A and B; S2 and S3 Figs.). In HT1080 fibrosarcoma cells, the period from one peak to the next was within 100 min in almost all cases. Therefore, the appropriate total observation time to detect the migration characteristics of HT1080 cells was regarded as 100 min or longer. The distribution of migration speed for HT1080 cells (n = 50) is shown in Fig. 5C. The unimodal distribution, as opposed to a bimodal one, indicates that the high- and low-speed values in the HT1080 two-speed mode are not fixed for the entire population but vary on a cell-to-cell basis within the population. The distribution of migration speed for 3T3-Swiss fibroblasts (n = 50) is shown in Fig. 5D. Compared with HT1080 fibrosarcoma cells, the 3T3-Swiss fibroblast population displayed a

HT1080 fibrosarcoma

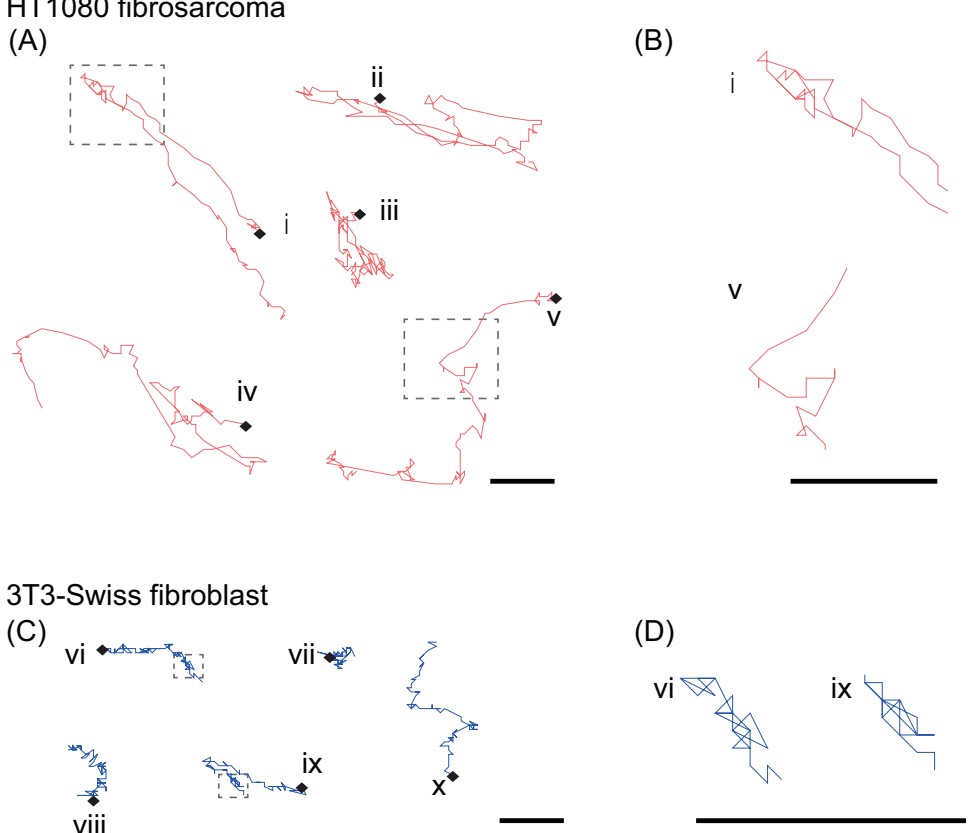

3T3-Swiss fibroblast

**Fig. 4. Automatically detected cell trajectories. (A, B)** HT1080 cells over 100 min. Trajectories with two-speed modes and an average migration speed of 0.02 μm/s or high are shown as typical. Enlarged images in the dashed boxed region in **(A)**-i and -v are shown in the upper and bottom sections of **(B)**, respectively. **(C, D)** 3T3-Swiss fibroblasts in 100 min. Trajectories with an average migration speed of 0.015 μm/s or low are shown as those typical. The enlarged images in the boxed region in **(C)**-vi and -ix are respectively shown in the left and right parts in **(D)**. Starting points are indicated with black rhombuses. Scale bars: 10 μm.

narrower distribution with a sharp peak at 0.015 μm/s. These findings indicate that HT1080 migration, at the population level, is characterized by heterogeneous migration speed with a broader unimodal distribution compared to the 3T3-Swiss population, rather than a biphasic distribution.

Fig. 5E and F and S4 and S5 Figs. show the time course of the turn angles in HT1080 cells and 3T3-Swiss fibroblasts. HT1080 cells showed a wide range of turn angles (Fig. 5E, S4 Fig.), whereas 3T3-Swiss fibroblasts showed lower turn angles (Fig. 5F, S5 Fig.) compared to HT1080 cells. Histograms showing the distribution of the turn angles indicated that HT1080 cells were uniformly distributed from 0° to 180° (Fig. 5G), whereas 3T3-Swiss fibroblasts were concentrated at an angle close to 0° (Fig. 5H). In the histograms of turn angles, the probabilities were high within the ranges of 0–20°, 40–50°, 90–100°, 130–140°, and 170–180° for both HT1080 cells and 3T3-Swiss fibroblasts (Fig. 5G and H). This pattern is attributed to the significant figures used in centroid calculations. Specifically, the analysis was performed based on centroid data calculated in pixels (0.65 μm/pixel). The histograms reflect the effect of discretizing centroid displacement, where cell centroid movements per unit time are limited to one of the adjacent pixels, resulting in peaks at approximate angles of 0°, 45°, 90°, 135°, and 180°.

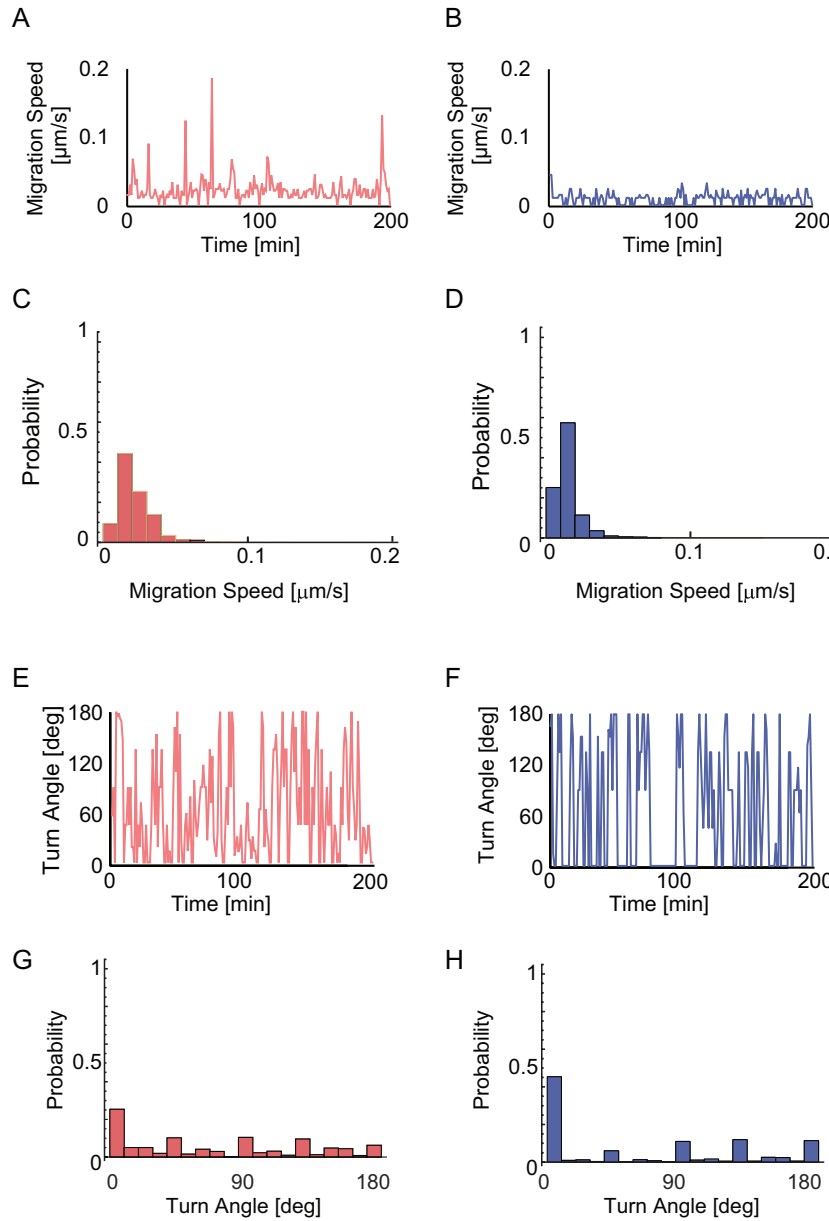

**Fig. 5. Quantitative characterization of migration trajectories.** The results for HT1080 cells **(A, C, E, G)** and 3T3-Swiss fibroblasts **(B, D, F, H)** are shown. **(A, B)** Time courses of the cell migration speed. **(C, D)** Stochastic distributions of the migration speed. Outliers larger than 0.2, specifically **(C)** **n** = 1 within 0.24–0.26, and **(D) n** = 1, 1, and 1 for each within 0.20–0.22, 0.22–0.24, and 0.38–0.40, are not shown in the histograms. **(E, F)** Time courses of the turn angle. **(G, H)** Stochastic distributions of the turn angle. **(C, D, G, H)** Data from **n** = 50 cells **(C, G:** 7398 time points in HT1080 fibrosarcoma cells; **D, H:** 12427 time points in 3T3-Swiss fibroblasts), which were applied to the discrimination analysis in Fig. 8, are summarized.

The motility parameters listed in the S1 Table were calculated from the cell-tracking data. The average values of these calculated parameters are presented in Fig. 6. The mean migration speed (Fig. 6A) of HT1080 cells was approximately 1.5 times higher than that of 3T3-Swiss fibroblasts (also 1.5 times in the median value, S6 Fig.). Differences were also observed in the "frequency of turns below thresholds," with the relative difference increasing as the threshold

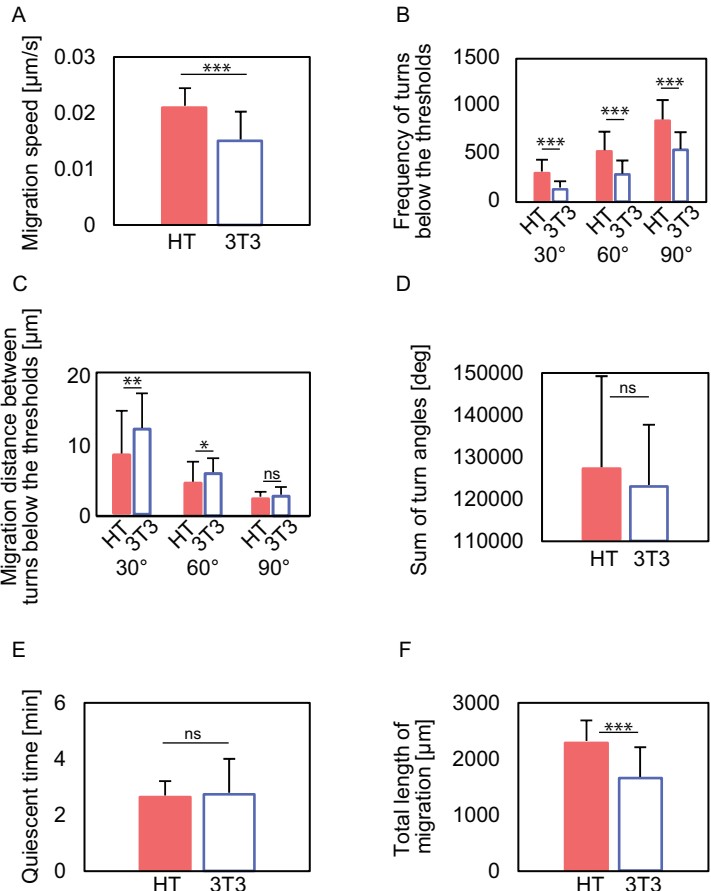

**Fig. 6. Quantitative characterization at the cell population level of migration trajectories.** Statistics of cell motility parameters extracted from the data obtained using automatic cell tracking. **(A)** Migration speed. **(B)** Frequency of turns below the thresholds of 90°, 60°, and 30°. **(C)** Migration distance between turns below the thresholds of 90°, 60°, and 30°. **(D)** The sum of turn angles. **(E)** Quiescent time. **(F)** Total migration length. The abbreviations HT and 3T3 indicate HT1080 cells and 3T3-Swiss fibroblasts, respectively. Data are mean + SD for n = 50 HT1080 cells and **n** = 50 3T3-Swiss fibroblasts used for the discrimination analysis in Fig. 8. $P$-values were calculated using the Welch t-test: * $P$ < 0.05, **$P$ < 0.01, ***$P$ < 0.001.

decreased (Fig. 6B). The "frequency of turns below 90°" in HT1080 cells was approximately 1.6 times higher than that in 3T3-Swiss fibroblasts. The difference was more than twice as high when the threshold was set at 30°. Differences were also observed in "migration distance between turns below thresholds" (Fig. 6C), where 3T3-Swiss fibroblasts migrated a longer distance to make a sharp change of direction compared to HT1080 cells. In contrast, no significant differences were found between the cell types in the "sum of turn angles" (Fig. 6D) and "time to stop" (Fig. 6E). The "quiescent time" refers to the duration during which a cell maintains a speed lower than a defined threshold (average speed) and was proposed as a potential parameter to differentiate the two-speed mode of HT1080 fibrosarcoma cells from the low-speed maintenance of 3T3-Swiss fibroblasts. However, contrary to expectations, the difference between these two behaviors was not reflected in the average quiescent time values. The "total length of migration" in HT1080 cells was approximately 1.4 times higher than that in 3T3-Swiss fibroblasts (Fig. 6F), which was almost the same as that for the migration speed (Fig. 6A).

Previous research has suggested that migration speed and turn angle are closely correlated [32,33]. To explore a more distinctive difference between a cancer cell model, HT1080 fibrosarcoma cell, and a non-malignant 3T3-Swiss fibroblast, the correlation coefficients between each motility parameter were calculated. The correlation coefficients for all combinations are summarized as heat maps in Fig. 7. A positive correlation was found between "migration speed" and "frequency of turns below thresholds" for both cell types. The most remarkable differences in correlation coefficients between the two cell types were found in combinations regarding the "sum of turn angles." Although the "sum of turn angles" alone exhibited large variance and no significant difference between HT1080 fibrosarcoma cells and non-cancer 3T3-Swiss fibroblasts (Fig. 6D), its correlation with "migration speed" and "frequency of turns below thresholds" differed between these cell types. In "migration speed" and "sum of turn angles," HT1080 fibrosarcoma cells showed a weak negative correlation, whereas 3T3-Swiss fibroblasts showed a positive correlation. In "frequency of turns below thresholds" and "sum of turn angles," HT1080 cells showed a strong negative correlation, whereas 3T3-Swiss fibroblasts showed almost no correlation. In addition, HT1080 cells showed a positive correlation of "migration distance between turns below thresholds" for each threshold and "sum of turn angles," while 3T3-Swiss fibroblasts showed a weaker correlation. These relationships cannot be extracted by simply comparing a single value. Collectively, the "sum of turn angles," which is one of the parameters that indicate the overall polarity change in the cell trajectory over a timescale of several tens of hours, is differently correlated to the motility parameters that define a shorter timescale and more local features of the trajectory or infinitesimal centroid displacements associated with cell morphological changes. The result implies different machinery in the multiscale regulation mechanism of cell migration among HT1080 fibrosarcoma cells and non-malignant 3T3-Swiss fibroblasts. Negative correlations were observed between "migration speed" and "migration distance between turns below thresholds" (thresholds: 30° and 60°) for HT1080 cells, whereas no correlations were identified for 3T3-Swiss fibroblasts. In terms of the relationship between migration speed and directionality, HT1080 fibrosarcoma cells tended to migrate directionally at higher migration speeds, whereas 3T3-Swiss fibroblasts showed no such correlation.

## Discrimination between cancer and normal cell models using LDA and QDA

In the cell trajectories obtained through automatic tracking, HT1080 cells demonstrated high migratory activity and a preference for directional migration. In contrast, 3T3-Swiss fibroblasts showed low migratory activity and a tendency to remain stationary. The differences in migration trajectories were well reflected in the motility parameters (S1 Table) extracted from the cell trajectories (Figs. 5 and 6). Additionally, correlation analysis clarified that correlation coefficients between the "sum of turn angles" and each parameter discriminated the characteristics of both cell types (Fig. 7). To evaluate the potential for cancer cell discrimination with a combination of motility parameters, LDA and QDA were applied to obtain discrimination boundaries, then sensitivity, specificity, and accuracy were estimated. Table 3 shows parameter sets with the top three highest accuracies. The parameter sets that included the "sum of turn angles" showed high accuracy, as predicted by the correlation analysis results in Fig. 7. QDA produced more accurate discriminant models than LDA, with a maximum sensitivity, or true positive rate, of 94.0%. Both LDA and QDA showed accuracies greater than 80%, with the parameter set yielding the highest accuracy identical to QDA. Those parameter sets consisted of "frequency of turns below 30°" and "sum of turn angles," presumed from the results of the correlation analysis in Fig. 7.

| HT1080 (N=50) | Migration Speed | Freq. 90° | Freq. 60° | Freq. 30° | MD. 90° | MD. 60° | MD. 30° | Sum Angles | Quiescent Time | Total L. |
|---|---|---|---|---|---|---|---|---|---|---|
| Migration Speed | 1.00 | 0.50 | 0.57 | 0.60 | 0.13 | −0.20 | −0.24 | −0.21 | 0.29 | 1.00 |
| Freq. 90° | 0.50 | 1.00 | 0.96 | 0.92 | −0.75 | −0.75 | −0.74 | −0.83 | 0.06 | 0.50 |
| Freq. 60° | 0.57 | 0.96 | 1.00 | 0.94 | −0.65 | −0.72 | −0.70 | −0.80 | 0.11 | 0.57 |
| Freq. 30° | 0.60 | 0.92 | 0.94 | 1.00 | −0.57 | −0.64 | −0.68 | −0.79 | 0.09 | 0.60 |
| MD. 90° | 0.13 | −0.75 | −0.65 | −0.57 | 1.00 | 0.84 | 0.80 | 0.76 | 0.12 | 0.13 |
| MD. 60° | −0.20 | −0.75 | −0.72 | −0.64 | 0.84 | 1.00 | 0.97 | 0.66 | 0.01 | −0.20 |
| MD. 30° | −0.24 | −0.74 | −0.70 | −0.68 | 0.80 | 0.97 | 1.00 | 0.64 | −0.01 | −0.25 |
| Sum Angles | −0.21 | −0.83 | −0.80 | −0.79 | 0.76 | 0.66 | 0.64 | 1.00 | 0.04 | −0.21 |
| Quiescent time | 0.29 | 0.06 | 0.11 | 0.09 | 0.12 | 0.01 | −0.01 | 0.04 | 1.00 | 0.29 |
| Total L. | 1.00 | 0.50 | 0.57 | 0.60 | 0.13 | −0.20 | −0.25 | −0.21 | 0.29 | 1.00 |

| Swiss3T3 (N=50) | Migration Speed | Freq. 90° | Freq. 60° | Freq. 30° | MD. 90° | MD. 60° | MD. 30° | Sum Angles | Quiescent Time | Total L. |
|---|---|---|---|---|---|---|---|---|---|---|
| Migration Speed | 1.00 | 0.61 | 0.65 | 0.60 | 0.26 | −0.05 | −0.08 | 0.50 | 0.35 | 1.00 |
| Freq. 90° | 0.61 | 1.00 | 0.94 | 0.90 | −0.47 | −0.71 | −0.53 | 0.01 | 0.10 | 0.61 |
| Freq. 60° | 0.65 | 0.94 | 1.00 | 0.92 | −0.33 | −0.70 | −0.53 | 0.04 | 0.10 | 0.65 |
| Freq. 30° | 0.60 | 0.90 | 0.92 | 1.00 | −0.30 | −0.64 | −0.66 | 0.06 | 0.07 | 0.60 |
| MD. 90° | 0.26 | −0.47 | −0.33 | −0.30 | 1.00 | 0.84 | 0.48 | 0.56 | 0.22 | 0.26 |
| MD. 60° | −0.05 | −0.71 | −0.70 | −0.64 | 0.84 | 1.00 | 0.65 | 0.37 | 0.14 | −0.05 |
| MD. 30° | −0.08 | −0.53 | −0.53 | −0.66 | 0.48 | 0.65 | 1.00 | 0.15 | 0.13 | −0.08 |
| Sum Angles | 0.50 | 0.01 | 0.04 | 0.06 | 0.56 | 0.37 | 0.15 | 1.00 | 0.18 | 0.49 |
| Quiescent time | 0.35 | 0.10 | 0.10 | 0.07 | 0.22 | 0.14 | 0.13 | 0.18 | 1.00 | 0.35 |
| Total L. | 1.00 | 0.61 | 0.65 | 0.60 | 0.26 | −0.05 | −0.08 | 0.49 | 0.35 | 1.00 |

-1.0      0      1.0

**Fig. 7. Correlation among motility parameters.** Correlation matrix representing correlation coefficients for 10 motility parameters for HT1080 cells (upper part) and 3T3-Swiss fibroblasts (lower part): migration speed, frequency of turns below the thresholds of 90° (Freq. 90°), 60° (Freq.

60°), and 30° (Freq. 30°), migration distance between turns below the thresholds of 90° (MD. 90°), 60° (MD. 60°), and 30° (MD. 30°), sum of turn angles (Sum Angles), quiescent time, and total migration length (Total L.). Correlation coefficients are indicated by numerical values and colored backgrounds.

**Table 3. List of motility parameter sets yielding top three accuracies.**

| Algorithm | 1st parameter | 2nd parameter | Sensitivity | Specificity | Accuracy |
|---|---|---|---|---|---|
| LDA | Frequency of turns below 30° | Sum of turn angles | 76.0% | 84.0% | 80.0% |
| | Migration distance between turns below 30° | Total migration length | 86.0% | 72.0% | 79.0% |
| | Frequency of turns below 60° | Sum of turn angles | 76.0% | 82.0% | 79.0% |
| QDA | Frequency of turns  below 30° | Sum of turn angles | 94.0% | 74.0% | 84.0% |
| | Frequency of turns  below 90° | Sum of turn angles | 90.0% | 76.0% | 83.0% |
| | Sum of turn angles | Total migration length | 88.0% | 78.0% | 83.0% |

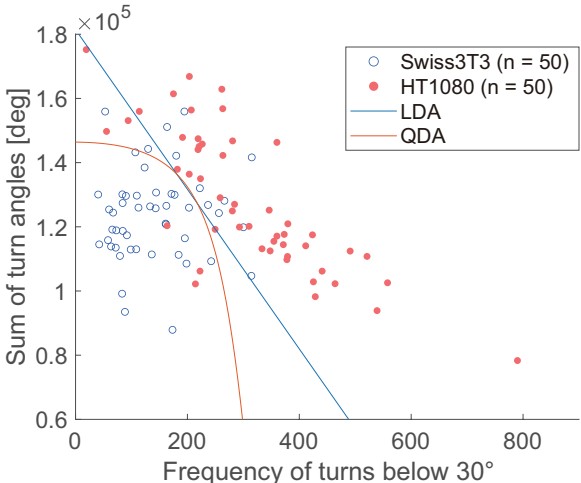

**Fig. 8. Discrimination curves of the result of the discrimination analysis for the dataset giving the best performance.** The parameters "frequency of turns below 30°" and "sum of turn angles" are plotted. The blue and orange lines represent the boundaries estimated by LDA and QDA, respectively. Their performances are listed in the first (LDA) and fourth (QDA) rows of Table 3.

Fig. 8 shows the discrimination curves calculated using LDA and QDA with parameters "frequency of turns below 30°" and "sum of turn angles." In the HT1080 cells, these parameters were negatively correlated and distributed in the upper region of the graph. In contrast, 3T3-Swiss fibroblasts showed no correlation, with the parameters distributed in the left-hand region, which contributed to the high performance of the discrimination. The effectiveness of the parameter set, consisting of "frequency of turns below 30°" and "sum of turn angles," was suggested for "frequency of turns below 30°" in the analysis of average values (Fig. 6B) and for the combination of "frequency of turns below 30°" and "sum of turn angles" in the correlation analysis (Fig. 7).

## Discussion

We developed a label-free automatic tracking method for phase contrast images, enabling quantitative characterization of cell motility phenotypes. We evaluated the potential application

of this method and demonstrated its usefulness as a novel tool for label-free discrimination of different types of mesenchymal mode migration. The feasibility of this approach was validated through its application to HT1080 fibrosarcoma and 3T3-Swiss fibroblast, both of which are known to exhibit mesenchymal mode migration when migrating as single cells on the surface of plastic tissue culture vessels [5]. The HT1080 fibrosarcoma cells and the 3T3-Swiss fibroblasts were discriminated with high accuracy, with label-free motility parameters, utilizing only a flat dish without a complex experimental system. Therefore, this method for cell discrimination and motility-based analysis is easy-to-use and inexpensive, enabling individuals without professional skills in image analysis—such as algorithm selection, pre- and post-processing, large training datasets, and special devices—to effectively engage with it. Our method may be particularly useful in early-phase high-throughput studies, such as target validation and preclinical drug development, where high throughput and low cost are required.

In this study, cell segmentation and tracking were applied to phase contrast microscopy. Since phase contrast images represent cells based on brightness values, cell segmentation using a Sobel filter was adopted. This enables the segmentation of cellular regions from the image and allows for the calculation of the center of the ROI. As the analysis in this study targeted single cells with no contact with other cells, the images were cropped to include only the single cells to be tracked (S1 and S2 Videos). This facilitates the linking of the centroid coordinates in the time-series images and enables smoothing using a convex hull. A typical issue with convex hulls, which provide smooth cell contours, is their inability to extract constricted cell contours, leading to the recognition of two or more cells as one (S1 Fig.). In contrast, this segmentation approach is suitable for motility analysis on the cellular level under culture conditions with no cell-cell contact. The detrimental effect of the smoothing process on the results of the cell motility analysis, such as distinguishing different mesenchymal mode migrations, is considered negligible. Rather, smoothing with the convex hull reduces noise in the discrimination analysis. A time interval of 1 min was selected to capture changes in centroids associated with morphological changes at the cellular level, as well as cell centroid displacement while excluding noise at the subcellular level. Collectively, cell segmentation and tracking at a spatiotemporal resolution focuso on cell body displacements and morphological changes at the cellular level were sufficiently accurate for discrimination based on cell motility, as demonstrated in the discrimination analysis of HT1080 fibrosarcoma cells and non-malignant 3T3-Swiss fibroblasts with high sensitivity, specificity, and accuracy (Table 3).

In the cell trajectories obtained by automatic tracking, HT1080 cells showed high migratory activity and a preference for directional migration. In contrast, 3T3-Swiss fibroblasts showed low migratory activity and tended to remain stationary. Time-course data for individual HT1080 cells (Fig. 5A, S2 Fig.) demonstrated that the two-speed mode is a characteristic feature of invasive malignant HT1080 fibrosarcoma cells at the single-cell level. However, this two-speed mode was not representative at the population level (Fig. 5C) because the high- and low-speed values varied among individual HT1080 cells. Our correlation analysis revealed that the combination of "migration speed," "sum of turn angles," and "frequency of turns below the thresholds (30°, 60°, 90°)" effectively enhanced the ability to distinguish between cell types. The motility parameters highlighted both similarities and differences in the mesenchymal migration modes of cancerous HT1080 fibrosarcoma cells and non-cancerous 3T3-Swiss fibroblasts when exhibited on flat tissue culture polystyrene surfaces.

A similarity between cancerous- and non-cancerous mesenchymal type migrations is characterized by the positive correlations between the "frequency of turns below a threshold" (assessed at 30°, 60°, and 90°) and "migration speed" in both HT1080 and 3T3-Swiss cells. The "frequency of turns below the threshold" reflects the number of turns occurring without a change in front-back polarity. These positive correlations (Fig. 7) suggest that cell

displacements are achieved through cell deformation without altering front-back polarity. The underlying mechanism involves the mutual interaction between cell body displacement and morphological changes, a process recognized as the coordinated protrusion of the leading edge and retraction of the rear, spatiotemporally integrated through complex regulatory pathways [34].

A notable difference between cancerous fibrosarcoma-type and non-cancerous fibroblast-type mesenchymal migration modes is characterized by the "sum of turn angles" when analyzed alongside other parameters, such as "migration speed" and "frequency of turns below a threshold (30°, 60°, 90°)." We interpret the "sum of turn angles" as the cumulative degree of polarity changes caused by cell deformation. This parameter alone exhibited substantial variance within each population of HT1080 fibrosarcoma cells and 3T3-Swiss fibroblasts, showing no significant difference between the two. However, correlation analysis revealed that the "sum of turn angles," when combined with other motility parameters, effectively highlighted differences in the migration characteristics of HT1080 fibrosarcoma cells and 3T3-Swiss fibroblasts at the population level. This distinction is likely attributable to differences in the processes used by these cells to achieve effective cell body propulsion. The strong negative correlation between the "frequency of turns below the threshold (30°, 60°, 90°)" and the "sum of turn angles" in the HT1080 fibrosarcoma cell population (Fig. 7) suggests that active cell deformation while maintaining front-back polarity results in a lower cumulative degree of polarity change. Together with the negative correlation with "migration speed," the "sum of turn angles" in combination with the "frequency of turns below the threshold (30°, 60°, 90°)" reflects a distinguishing mechanism in HT1080 fibrosarcoma cells. In this mechanism, effective cell body propulsion is attained through cell deformation while preserving polarity. In contrast, in 3T3-Swiss fibroblast migration, the "sum of turn angles" showed a positive correlation with "migration speed," but no correlation with "frequency of turns below the threshold (30°, 60°, 90°)" (Fig. 7). This pattern suggests that the relationship between the "sum of turn angles," "frequency of turns below the threshold," and "migration speed" reflects a mechanism by which effective cell propulsion is achieved through cell deformation accompanied by polarity changes, regardless of their magnitude.

Another noteworthy migration characteristic demonstrated in the analysis is the heterogeneity of HT1080 fibrosarcoma cells. HT1080 cell migration was temporally heterogeneous, with transient peaks in migration speed (Fig. 5A). The fast-slow migration modes might be driven by similar mechanisms to those reported in fish keratocyte cells, where migration switches between fast and slow modes due to variations in intracellular diffusion rates and structural changes in the lamellipodium, transitioning between thin and thick morphologies [35]. In contrast, 3T3-Swiss fibroblasts showed almost no peaks and migrated around the mean value (Fig. 5B). A wider distribution in HT1080 fibrosarcoma cells compared to 3T3-Swiss fibroblasts (Fig. 5C and D) reflects the heterogeneity of cancer cells on a longer time scale and/or at the cell-to-cell level [36–38]. The findings suggest that our motility-based analysis has the potential for high-throughput analysis of heterogeneity in terms of migration characteristics and identifying regularities.

The method has the potential to cope with the cancer cell heterogeneity, which is a major challenge in cancer therapy [39,40]. The key to successfully identifying fine-grained differences in heterogeneous cell motility lies in the spatiotemporal resolution needed to capture both cell body displacements and polarity changes associated with morphological alterations. Accordingly, the requirements for distinguishing between the mesenchymal migration modes of cancerous fibrosarcoma and non-cancerous fibroblast cell types on flat tissue culture polystyrene are as follows: i) It is necessary to detect complete cell tracks at 1-min intervals for over 1 h, corresponding period from one peak to the next in two-speed mode migration

of cancerous HT1080 fibrosarcoma. However, prolonged image acquisition using fluorescence microscopy raises concerns about phototoxicity, emphasizing the need for label-free imaging methods. Thus, label-free segmentation and tracking present a suitable option. The method needs to minimize track fragmentation errors caused by spatiotemporal variations, such as fluctuations in image intensity and discontinuities in cell trajectories. ii) Additionally, the method needs to detect polarity changes resulting from cell deformation. This includes accurately capturing thin lamellipodia while remaining insensitive to subcellular structures such as the organelles. For instance, the method requires detecting small phase shifts originating exclusively from the lamellipodia, not from other intracellular fine structures. Our segmentation and tracking method meets these requirements, enabling the identification of fine-grained characteristics in the mesenchymal mode migration of cancerous fibrosarcoma and non-cancerous fibroblast cell types on flat tissue culture polystyrene surfaces.

Recently developed segmentation and tracking tools for label-free images [23,25] have the potential to successfully identify fine-grained differences in heterogeneous mesenchymal cell migration, provided they meet certain requirements. For applications focused solely on cell tracking, an effective strategy for phase contrast micrographs is to detect and track the nucleus as a reference point, as the nuclear region is clearly distinguishable in phase contrast images [27,28,41]. This strategy is robust even for cells migrating under high-density conditions with a number of cell–cell contacts. However, it is unable to extract the cell outline, which is essential for detecting polarity changes caused by cell deformation—a key feature for distinguishing between cancerous and non-cancerous mesenchymal mode migrations. Given images with adequate spatiotemporal resolution, such as the 1-min interval over 1 h used in this study, user-friendly tracking tools like TrackMate [42], combined with a robust segmentation algorithm such as CellPose [43,44], could be highly effective. Adjustments CellPose to detect cell outlines from phase contrast images with small phase shifts, while remaining insensitive to intracellular structures, would further increase its utility for such applications. Other deep learning-based tracking and segmentation packages, such as Usiigaci [29], can be used if properly trained with appropriate datasets. Our method and those of others that fulfil the requirements identified through the analysis (Figs. 5–8) can characterize and discriminate cancer and non-cancer cells based on mesenchymal migration on flat tissue culture polystyrene. Furthermore, state-of-the-art deep learning-based methods are expected to reduce manual processes, such as threshold adjustment in edge detection in our method (Fig. 1). Regarding the discrimination analysis, this study was limited to verifying the best accuracy by combining the "frequency of turns below 30º" and the "sum of turn angles" with QDA (Table 3, Fig. 8). Adopting three or more parameters is possible to further improve discrimination efficiency. Additionally, a support vector machine classifier would enhance the discrimination accuracy. Another consideration is that target cells are limited to those with no cell–cell contact. As the discrimination considers fine-grained differences in propulsive force generation associated with polarity changes due to single-cell deformation, the influence of cell–cell contacts on discrimination accuracy is significant. Exploring the effect of cell–cell contact is a topic for further research.

Cell migration is a highly studied area because it is related to various biochemical processes driven by complex interactions among numerous components, such as cell adhesion proteins, receptor proteins, the actin cytoskeleton, and mitochondria [11,45,46]. In cell motility studies incorporating quantitative analysis of cell migration trajectories over long periods, label-free approaches are preferred due to minimized phototoxicity. Manual tracking or manually trained processes remain the preferred options for quantitative characterization of cell migration from label-free images, such as phase contrast micrographs [15,47–49]. The proposed automated cell motility tracking tool further facilitates efficient and accurate analysis of cell

motility, specifically identifying fine-grained characteristics of heterogeneous mesenchymal mode migration in cancerous and non-cancerous mesenchymal mode migrations on flat tissue culture polystyrene, without the need for time-consuming work. Our method enables the determination of cellular states by linking cell motility to these functions. Expanding the label-free method to characterize cell migration, including the underlying mechanism of propulsive force generation, for a variety of cell types is an important consideration for future research. As the flexibility and ease of use of this method are harnessed to facilitate better analysis across a wide range of cell types, biological knowledge linking cell motility and function will continue to grow. In biomedical applications, our label-free cell tracking and motility phenotyping approach, combined with databases on cell motility and function, can be used in fields requiring high-throughput and an emphasis on cell-to-cell variability. This will be particularly useful for anti-cancer drug discovery and tailored therapy [50,51], as validated by the cancer cell discrimination achieved in this study.

## Supporting Information

**S1 Fig.  Typical cases of stop tracking.**  The process did not work when a cell exhibited a constricted shape due to failure to trace the cell outline during the smoothing process (A). Another case is when cell-cell contacts occurred during the cell tracking. The cells (red outlines) were recognized as a single cell as indicated with the white outline (B).
(PDF)

**S2 Fig.  Time courses of migration speed of HT1080 cells from automatically detected cell trajectories.**  The data were obtained from the same trajectories used in the discriminant analysis.
(PDF)

**S3 Fig.  Time courses of migration speed of 3T3-Swiss fibroblasts from automatically detected cell trajectories.** The data were obtained from the same trajectories used in the discriminant analysis.
(PDF)

**S4 Fig.  Time courses of turn angle from automatically detected cell trajectories.** The data were obtained from the same trajectories used in the discriminant analysis.
(PDF)

**S5 Fig.  Time courses of turn angle from automatically detected cell trajectories.**  The data were obtained from the same trajectories used in the discrimination analysis.
(PDF)

**S6 Fig.  Stochastic distributions of the migration speed and turn angle. The line in the box is the median.** The whiskers represent the lower and upper extremes. Outliers are indicated as single dots. The data of n = 50 HT1080 fibrosarcoma cells (n = 7398 time points) and n = 50 3T3-Swiss fibroblasts (n = 12427 time points) used in the discrimination analysis in Fig. 8 are summarized. (A) Migration speed. Outliers larger than 0.2, specifically n = 1 within 0.24–0.26 in HT1080, and n = 1, 1, and 1 for each within 0.20–0.22, 0.22–0.24, and 0.38–0.40 in 3T3-Swiss are not shown. (B) Turn angle.
(PDF)

**S1 Table.  List of motility parameters.** The parameters were calculated from the cell-tracking data and used in both linear discriminant analysis (LDA) and quadratic discriminant analysis (QDA) for cell-type discrimination.
(DOCX)

**S1 Video. Representative result of automatic label-free tracking for an HT1080 cell.** Fast-forward video depicting a total time of 180 min. The white outline indicates the automatically detected cell outline. The automatically tracked cell centroid at each time point is captured at the center of the image (cropped ROI).
(AVI)

**S2 Video. Representative result of automatic label-free tracking for a 3T3-Swiss fibroblast.** Fast-forward video of a total duration of 180 min. The white outline indicates the automatically detected cell boundary. The automatically tracked cell centroid at each time point was captured at the center of the image (cropped ROI).
(AVI)

## Acknowledgments

N/A

## Author contributions

**Conceptualization:** Hiromi Miyoshi.

**Data curation:** Sota Endo.

**Formal analysis:** Sota Endo.

**Investigation:** Sota Endo, Shotaro Yamamoto.

**Methodology:** Hiromi Miyoshi.

**Project administration:** Hiromi Miyoshi.

**Software:** Sota Endo.

**Supervision:** Hiromi Miyoshi.

**Writing – original draft:** Sota Endo, Shotaro Yamamoto.

**Writing – review & editing:** Hiromi Miyoshi.

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
