## [Decision Letter · Decision Letter 0]

2 Aug 2024

PONE-D-24-28597Development of Label-Free Cell Tracking for Cell Discriminant Analysis Based on Cell MotilityPLOS ONE

Dear Dr. Miyoshi,

Thank you for submitting your manuscript to PLOS ONE. After careful consideration, we feel that it has merit but does not fully meet PLOS ONE’s publication criteria as it currently stands. Therefore, we invite you to submit a revised version of the manuscript that addresses the points raised during the review process.

We look forward to receiving your revised manuscript.

Kind regards,

Ruo Wang

Academic Editor

PLOS ONE

Journal Requirements:

"This work was supported by Tokyo Metropolitan Government Advanced Research Grant Number R2-2, and TMU strategic research fund for social engagement."

Please note that funding information should not appear in the Acknowledgments section or other areas of your manuscript. We will only publish funding information present in the Funding Statement section of the online submission form. Please remove any funding-related text from the manuscript.

3. We are unable to open your Supporting Information files:

S1_Fig.epsS2_Fig.eps

Please kindly revise as necessary and re-upload.

Reviewers' comments:

Reviewer's Responses to Questions

**Comments to the Author**

1. Is the manuscript technically sound, and do the data support the conclusions?

Reviewer #1: Partly

Reviewer #2: Partly

Reviewer #3: Partly

Reviewer #4: No

2. Has the statistical analysis been performed appropriately and rigorously? 

Reviewer #1: No

Reviewer #2: Yes

Reviewer #3: Yes

Reviewer #4: No

3. Have the authors made all data underlying the findings in their manuscript fully available?

Reviewer #1: No

Reviewer #2: No

Reviewer #3: Yes

Reviewer #4: Yes

4. Is the manuscript presented in an intelligible fashion and written in standard English?

Reviewer #1: Yes

Reviewer #2: No

Reviewer #3: Yes

Reviewer #4: No

5. Review Comments to the Author

Reviewer #1: In this paper, the authors developed an automated label-free cell tracking system and compared the motility parameters of cancer and non-cancer cells. They showed that the two cell types can be discriminated by parameters related to turning and argued the usefulness of their high throughput methodology. Figures 5 and 6 summarize multiple motility parameters extracted from the individual tracking data, and Figures 7 and 8 highlight parameters to discriminate cancer and non-cancer cells. Though limited to the two cell types, the methods may contribute to the basic research field. The authors have appeared to validate the method regarding accuracy, sensitivity, and others; however, some concerns remain as listed below.

1) The problems of ref.17 is described in the introduction, but the reference also developed label-free tracking applicable for phase-contrast images. Besides this, as perhaps the authors have known, various automated, label-free cell tracking methods for mammalian cells (e.g., DOI:10.1016/j.isci.2022.104678, 10.1109/ICMLA55696.2022.00057, 10.1038/s41598-023-50227-9) and microorganisms (e.g., DOI: 10.1038/s41598-022-27158-y) have been published. The authors should cite some related papers and argue the advantages of the method developed in this study more carefully.

2) “high throughput” is likely one of the merits of the present method. However, 20 or less cells were measured in Fig 5; measured cell numbers in Fig 6 are unclear. On the other hand, Fig 8 is a summary of 50 cells. The authors should solve the inconsistency of the sample size or carefully explain why they submitted the data from such a small sample size because the gap in the sample size between experiments could make statistical data interpretation difficult.

3) Figs 5 and 6 show faster speed of cancer cells than non-cancer cells. The high speed in cancer cells would be due to the spike-like speed increment seen in Fig 5c. Indeed, the transient high speed is distinguished from non-cancer cells, but the cancer cells seem to move at a similar speed as non-cancer cells most of the observation time (Figs C and D show the same peak speed). Isn’t it reasonable that the transition between the two-speed modes (major low speed and minor high speed) is the speed property unique to the cancer cell? Please explain (i) why the authors focused on the mean speed and (ii) discuss the possible cause of the high-speed locomotion.

Reviewer #2: The authors proposed a method for automatic cell tracking in phase-contrast microscopy images. In detail, they proposed a segmentation process using imaging tools of MATLAB and identified quantitative measures to characterize the migration trajectories. Several tests were performed on fibrosarcoma cells as a cancer mass as well as on normal cell models to validate the proposed method.

In detail, the segmentation process performed with MATLAB does not consider the features of microscopy phase-contrast images (halo, low contrast, and several other imaging artifacts).

Segmentation and cellular tracking are the pillars of every microscopy image analysis procedure, and the literature on phase-contrast imaging is rich. However, the manuscript does not compare any state-of-the-art methods expecially in the artificial intelligence field. On the other hand, the proposed method has not addressed the main issues in phase-contrast images. For example, the cells have low contrast concerning the background, complex and different structures varying in time, and touch or overlap. A recent survey can be found here: https://doi.org/10.3390/a15090313

The quantitative analysis to characterize the phenotypes should be applied to more cell lines for validation.

In my opinion, the manuscriopt appears in an early stage and requires significant improvements to be ready for publication.

Reviewer #3: The authors present an automatic tracking method for phase-contrast images of fibroblast cells, which works label-free and is capable to discriminate different cell lines according to motility Parameters, which are used as input for linear and quadratic classification methods. The method is interesting, while the findings of the authors should be clarified in more detail.

In line 66 on page 11 (Introduction). The authors Mention 3t3-Swiss fibroblasts. The name of the cell line was differently written in the abstract of the presented work (line 22 of page 9).

In line 78 on page 11. the letter 'phi' for the Diameter of a polystyrene dish is confusing?

Authors mention an automatic tracking tool, but use a manually adjusted threshold Adjustment (line 101 on page 12)?

Fig 1B, It is not clear to the reader how the image processing from 'Fill' to 'Erode' reduce in dimension?

In Addition. The 'original' and the 'convex hull' images are very different, why?

In line 176 on page 16 (Results). The authors mention that the routine stops when a cell Encounter another one. How this event is detected automatically? by the area of the cell?

What is the average size and area of the investigated cells?

Authors mention 5 typical trajectories. These ones have been chosen with which criteria? (line 196 on page 17)

In line 205 on page 18 the authors mention a lower Migration speed of 3t3-Swiss cells. This value is the median value? Which parameter was considerate to see if these 2 datasets are significantly different?

In Fig. 5c & d a box chart plot would maybe better represent the shown data.

In line 220 on page 18 the authors mention the pixel resolution. What is the pixel resolution of this experiment?

Why the authors use only 2 points to calculate the turning angle? More datapoints could give a more consistent turning angle? The choice of 2 data-points for the angle calculation is due to the characteristic time of a turning angle? What is the error of such a measurement?

In line 234 on page 19. the authors mention a 1.5 times higher migration speed? How it looks for the median value?

Authors mention no differences for 'sum of turns', but later on in line 256 on page 20 mention the most remarkable difference for 'sum of turns'. The 'sum of turn angle' has a quite big variance. I do not understand the physical meaning of this parameter (sum of turn angle). The authors could provide more information.

Why the authors limited their classification to 2 classifiers? A Support Vector Machine classifier would maybe provide a better outcome?

Reviewer #4: The authors propose a method for tracking cells present in phase contrast

images and performed discrimination of cell types based on trajectory

characteristics with potential applications in cancer, tissue healing, and

other areas.

The authors state that "Cell phenotyping based on static data **should be**

complemented by dynamic data extracted from time-dependent cell

characteristics" but this depends on the study and application. Maybe a better

assertion is "For some studies it is important that cell phenotyping based on

static data ** be ** complemented by dynamic data extracted from time-dependent

cell characteristics". Please revise.

"Additionally, if it is a deep-learning strategy, huge training data is

indispensable". This is a misleading assumption. Pre-trained models have come a

long way and they can be effectively fine tuned using a handful of new

annotations (new ground truth) to render excellent segmentation solutions on

new datasets at hand. It might not take more than a day or so to annotate a

small set of new images with hundreds of cells to obtain a successful fine

tuned model for your data. Readily available tools like Cellpose, StarDist,

Omnipose are popular and proven ones adopted by many. They have been used in

many image categories including phase contrast images. The authors need to

compare their methods with results from these developments to justify proposing

their new segmentation strategy.

Deep learning based methods have dominated in cell segmentation tasks - see,

for example, top ranking segmentation methods in the publication cited by the

authors "The Cell Tracking Challenge: 10 years of objective benchmarking", more

specifically the Cell Segmentation Benchmark. Such evidence, and others,

greatly weakens the development of cell segmentation schemes using classical

methods (Sobel filter, morphological dilation and erosion, thresholding), as it

is done by the authors. In special, manually adjusting thresholds ("...manually

adjusted [threshold] for each cell type") is not favorable and quite frankly

not needed as the methods mentioned above eliminate such step. My strong

suggestion is for the authors to adopt these other more robust and proven

methods to obtain the segmentations they need and eliminate their proposed

method, which might not work, for example, for close proximity cells.

The literature on image segmentation is very poor and does not reflect the

state of the art. There is a single reference to a review article [26] but none

that actually presents contemporary cell segmentation methods. The authors must

update the literature.

The literature is also very poor on tracking methods and there is no

justification why not adopting other popular methods readily available,

e.g. TrackMate, in tools like ImageJ/Fiji. While it might personally be a

valuable algorithmic experience, the proposed tracking method does not offer

any new novelty to be considered for publication at Plos ONE.

"The tracking was stopped when cell division or contact with other cells

occurred". How is this done and computed? It is not clear how this step is

achieved. Please elaborate so readers are able to replicate your method.

Similarly, how does the method decide if segmentation mistakenly generated a

single cell, as mentioned in "Moreover, if cell-cell contacts occurred during

cell tracking, this method recognized the cells as a single cell (S1B Fig.). In

these cases, cell tracking was stopped at that point and the tracking data

before contact were used for analysis"? Please explain.

Typo: "The parameters *are* affected depending on the total tracking time".

Please explain the conversion mentioned in "We converted the parameters

into those that were tracked for 30 h".

What do LDA and QDA stand for? Linear and quadratic discriminant analysis? If

so, please make it explicit so one can readily understand these acronyms.

"HT1080 cells was transient increases in the migration speed (Figs. 5A; S2A

Fig.)". Please rephrase, it is not understandable. Same for the following, "In

both cell types, the peaks at 0<b0>, 45<b0>, 90<b0>, 135<b0>, and 180<b0> are existed. This is

due to pixel resolution where the cell centroid migrated to one of the adjacent

pixels, the turn angles were discrete value of 0<b0>, 45<b0>, 90<b0>, 135<b0>, or 180<b0>".

Typo: "*Calculation* of cell centroids". And many other typos throughout the

manuscript. Please revise and correct grammar as well.

The authors state in their conclusion that "Therefore, this method for cell

discrimination and motility-based analysis of the cell status is easy to use

and inexpensive enabling anyone without professional skills of image analysis,

such as algorithm selection and pre- and post-processing, huge training data

and special devices. Furthermore, it is highly versatile and can be applied in

basic cell biological research and biomedical applications", but there is no

evidence of usage in different image categories, different cell morphologies

and sizes, and crowded situations where tracking is more challenging.

Unfortunately, the authors show a considerable lack of maturity and poor

investigation of availability of computing methods to tackle the image

processing problem.

The authors should probably concentrate their contribution on the discriminant

analysis beyond the two datasets they present in the manuscript. I am not

familiar with the quantitative characterization of cell motilities in different

species but hopefully what the authors have achieved in this specific point can

be valuable and extended to other types of cells besides HT1080 and 3T3-Swiss.</b0></b0></b0></b0></b0></b0></b0></b0></b0></b0>

6. PLOS authors have the option to publish the peer review history of their article (what does this mean? ). If published, this will include your full peer review and any attached files.

**Do you want your identity to be public for this peer review?** For information about this choice, including consent withdrawal, please see our Privacy Policy .

Reviewer #1: No

Reviewer #2: No

Reviewer #3: **Yes: ** David Dannhauser

Reviewer #4: No

---

## [Author Response · Author response to Decision Letter 0]

16 Jan 2025

This is a brief version due to a word limit for the Respond to Reviewers field in the submission system. If necessary, please see also the full version “Response to Reviewer” part (the same thing except for including some quotes from the revised manuscript) in the attached revised manuscript file.

The English grammar and stylistics of our manuscript have been revised by a native English proofreader. in response to the comment from one of the reviewers. These modifications did not affect the conclusions.

*****

Response to Reviewer #1

Reviewer #1: In this paper, the authors developed an automated label-free cell tracking system and compared the motility parameters of cancer and non-cancer cells. They showed that the two cell types can be discriminated by parameters related to turning and argued the usefulness of their high throughput methodology. Figures 5 and 6 summarize multiple motility parameters extracted from the individual tracking data, and Figures 7 and 8 highlight parameters to discriminate cancer and non-cancer cells. Though limited to the two cell types, the methods may contribute to the basic research field. The authors have appeared to validate the method regarding accuracy, sensitivity, and others; however, some concerns remain as listed below.

Response:

We are grateful to Reviewer #1 for the critical comments and useful suggestions that have helped us improve our paper considerably. As indicated in the following responses, we have taken all these comments and suggestions into account. In the revised manuscript, all amendments except for minor corrections in English grammar and stylistic have been indicated in blue.

1.1 The problems of ref.17 is described in the introduction, but the reference also developed label-free tracking applicable for phase-contrast images. Besides this, as perhaps the authors have known, various automated, label-free cell tracking methods for mammalian cells (e.g., DOI:10.1016/j.isci.2022.104678, 10.1109/ICMLA55696.2022.00057, 10.1038/s41598-023-50227-9) and microorganisms (e.g., DOI: 10.1038/s41598-022-27158-y) have been published. The authors should cite some related papers and argue the advantages of the method developed in this study more carefully.

Response 1.1

We wish to express our deep appreciation to the Reviewer for the insightful comment. As the Reviewer pointed out and we also knew, there have been various automated, label-free cell tracking methods. Upon careful examination of the advantages of the method developed in this study compared with related papers, together with the revisions according to other comments from the Reviewer, we specified the advantage of our method as characterization and discrimination of the different mesenchymal modes of movement exhibited by cancer and non-cancer cells on a polystyrene dish. The argument about the advantages of the methods has been moved to the Discussion in the revised version. In the Introduction, related papers providing background to understand the discussion on these advantages have been cited. The amendments to the Introduction and the Discussion are as follows.

Introduction:

- The papers reporting label-free phenotyping cells during lineage specification ([9] Kimmel et al., IEEE/ACM Trans. Comput. Biol. Bioinform., 2021; [10] Buggenthin, Nat. Methods, 2017) have been cited to explain that cell motility serves as a good indicator to characterize the cellular state (lines 51–54, p. 4).

- The papers reporting tracking of cells in phase contrast micrographs ([26] Han et al., IEEE ICMLA, 2022; [27] Holme et al., Sci. Rep., 2023; [28] Gu et al., iScience, 2022; [29] Tsai et al., SoftwareX, 2019; [30] Al-Zaben et al., Sci. Rep., 2019, ref 17 in the original version) have been cited (lines 76–84, p. 5) with the description of the strengths and limitations of microscopy for label-free and fluorescently labeled cells. Here, the problem in ref. 30 (ref. 17 in the original version) pointed out by the Reviewer has been solved.

The argument on the advantages of our method in the Discussion:

The distinguishing point of this study is the successful characterization of different types of mesenchymal modes, specifically a malignant invasive cancer-type and non-malignant fibroblast-type migration, of cells cultured on plastic tissue culture vessel surfaces (lines 399–403, p. 24). We have also provided a basis for discrimination, i.e., the reason why the “sum of turn angles” was a key parameter to highlight the difference in the migration characteristics of HT1080 fibrosarcoma cells and 3T3-Swiss fibroblasts (lines 451–472, pp. 27–28). The discrimination between two types of mesenchymal mode migration with motile parameters extracted from a label-free image and its basis in relation to the fine-grained difference in the scheme of propulsive force generation is, as far as we know, a first.

In the 8th paragraph of the Discussion (lines 503–521, pp. 29–30), the proposed method and state-of-the-art methods have been evaluated in the context of the requirements of segmentation and tracking to identify the fine-grained characteristics in the cancer type and the non-cancer fibroblast-type mesenchymal migration on flat tissue culture polystyrene, which is the greatest strength of our method. In the paragraph, another review article ([25] Maddalena et al., Algorithms, 2022) is cited in addition to the review articles ([23] Vicar et al., BMC Bioinformatics, 2019; [24] Maska et al, Nat. Methods, 2023) in the original version. Moreover, as specific methods for comparison, CellTraxx (for nucleus, [27] Holme et al. Sci. Rep., 2023), a UNet-based method (for nucleus, [28] Gu et al., iScience, 2022), Bright2Nuc (for nucleus, [41] Atwell et al., Cell Rep. Methods, 2023), TrackMate ([42] Tinevez et al., Methods, 2017; [43] Ershov et al., Nat. Methods, 2022), CellPose ([44] Pachitariu et al., Nat. Methods, 2022), and Usiigaci ([29] Tsai et al., SoftwareX, 2019) have been presented.

1.2 “high throughput” is likely one of the merits of the present method. However, 20 or less cells were measured in Fig 5; measured cell numbers in Fig 6 are unclear. On the other hand, Fig 8 is a summary of 50 cells. The authors should solve the inconsistency of the sample size or carefully explain why they submitted the data from such a small sample size because the gap in the sample size between experiments could make statistical data interpretation difficult.

Response 1.2:

We apologize for the confusion due to the inadequate data presentation. The inconsistency of the sample size in the original version was because the results in Figs. 5 and 6 were selected based on time-course data longer than 60 min, even though the migration trajectory data applied to the discriminant analysis included trajectories less than 60 min. To solve the inconsistency, Fig. 5 (C, D, G, H) and Fig. 6 are replaced with the results for the statistical analysis for all the trajectories applied to the discriminant analysis (Fig. 5: lines 296–298, pp. 17–18; Fig. 6: lines 323–325, p. 19).

1.3 Figs 5 and 6 show faster speed of cancer cells than non-cancer cells. The high speed in cancer cells would be due to the spike-like speed increment seen in Fig 5c. Indeed, the transient high speed is distinguished from non-cancer cells, but the cancer cells seem to move at a similar speed as non-cancer cells most of the observation time (Figs C and D show the same peak speed). Isn’t it reasonable that the transition between the two-speed modes (major low speed and minor high speed) is the speed property unique to the cancer cell? Please explain (i) why the authors focused on the mean speed and (ii) discuss the possible cause of the high-speed locomotion.

Response 1.3

We appreciate the Reviewer’s comment on this point. We agree that the transition between the two-speed modes (major low speed and minor high speed) is the speed property unique to the cancer cell. We had the same idea as the Reviewer during the preparation of the original version. The answers to the specific questions (i, ii) regarding this matter are as follows.

(i) Why do the authors focus on the mean speed?

Based on the idea that the transition between the two-speed modes of migration characterizes cancer cell migration, we created the histograms in Fig. 5C and D in the original version, expecting a bimodal distribution for cancer cells and a monomodal distribution for non-cancer cells. The result revealed that HT1080 migration on the cell population level was statistically characterized as heterogeneous “migration speed” with a broader unimodal distribution compared with the 3T3-Swiss population rather than a biphasic distribution.

We also tried to detect the two-speed mode with a parameter, “Quiescent Time,” which is the time duration maintaining a speed lower than a threshold (average speed) (Fig. 6E). However, the difference between cancer- and non-cancer cells was not significant, both in the original (the data longer than 300 min) and revised (the n = 50 data applied to the discrimination analysis) versions.

Taken together, as the Reviewer suggested, time-course data for each single HT1080 cell clearly showed that the two-speed mode is one of the features of a cancer cell at the single-cell level. However, at the cell population level, the statistical data demonstrated that mean speed was a parameter that characterizes cancer and non-cancer cells. Accordingly, in the revised version, the 2nd and 3rd paragraphs of “Quantitative characterization of motilities for HT1080 fibrosarcoma cells and 3T3-Swiss fibroblasts” have been amended to clearly show why we focused on the mean speed, which have been added as follows.

The 2nd paragraph:

- The description of the two-speed mode of single cell migration of HT1080 has been added (lines 265–267, p. 16).

- The description of the statistical characterization of HT1080 cell migration at the cell population level has been added (lines 270–273, p. 16).

The 4th paragraph:

- A description of the intention behind introducing the parameter “quiescent time” has been supplemented (lines 310–314, p. 18).

(ii) the possible cause of the high-speed locomotion

From a data analysis viewpoint, the possible cause of the averaged higher value in speed for the HT1080 cell population is that averaging the transient higher values and the low values almost the same as the 3T3-Swiss, yields a higher speed in HT1080 on average than in 3T3-Swiss. From a cell biological viewpoint, a possible cause is described in the Discussion (lines 475–478, p. 28).

***

Response to Reviewer #2

Reviewer #2: The authors proposed a method for automatic cell tracking in phase-contrast microscopy images. In detail, they proposed a segmentation process using imaging tools of MATLAB and identified quantitative measures to characterize the migration trajectories. Several tests were performed on fibrosarcoma cells as a cancer mass as well as on normal cell models to validate the proposed method.

Response:

We are grateful to Reviewer #2 for the critical comments and useful suggestions that have helped us improve our paper considerably. As indicated in the following responses, we have extensively revised the manuscript by considering all these comments and suggestions. In the revised manuscript, all amendments except for minor corrections in English grammar and stylistic have been indicated in blue. We would be glad to respond to any further questions and comments that the Reviewer may have.

2.1 In detail, the segmentation process performed with MATLAB does not consider the features of microscopy phase-contrast images (halo, low contrast, and several other imaging artifacts).

Response 2.1

We agree that we should consider the features of microscopy phase-contrast images (halo, low contrast, and several other imaging artifacts). Accordingly, first, in the 3rd paragraph of the Introduction, we presented the features usable to birefringent specimen containers (lines 74–76, p. 5), halo artifact for the thick specimen (lines 76–77, p. 5). Then, based on these features, we described the case-by-case points of consideration in the use of tracking tools for such phase-contrast images (lines 77–84, p. 5). The issue regarding the low contrast for the thin specimen has been addressed in the Introduction (lines 89–96, p. 5). We would like to note in particular that the case-by-case points have been presented to highlight points related to the novelty of this study, which is the characterization of the different mesenchymal modes of movement exhibited by cancer cells and non-cancer cells on a polystyrene dish.

2.2 Segmentation and cellular tracking are the pillars of every microscopy image analysis procedure, and the literature on phase-contrast imaging is rich. However, the manuscript does not compare any state-of-the-art methods expecially in the artificial intelligence field. On the other hand, the proposed method has not addressed the main issues in phase-contrast images. For example, the cells have low contrast concerning the background, complex and different structures varying in time, and touch or overlap. A recent survey can be found here: https://doi.org/10.3390/a15090313

Response 2.2

We agree that we should have compared the proposed method with the state-of-the-art methods, including the issues in phase-contrast images. Accordingly, we have revised the manuscript to include these points. The proposed method and state-of-the-art methods have been evaluated in the context of the requirements of segmentation and tracking to identify the fine-grained characteristics in the cancer type and non-cancer fibroblast-type mesenchymal migration on flat tissue culture polystyrene, which is the most significant aspect of our method. The points of the revision are as follows.

First, a description of the novelty of the research—the discrimination of two types of mesenchymal mode migration with motile parameters—has been added in the Discussion (lines 399–403, p. 24) with the background in the Introduction (lines 46–60, pp. 3–4). Additionally, the requirements of segmentation and tracking for the mesenchymal mode discrimination identified throughout the process from the image acquisition to the discrimination have been discussed in the Discussion (lines 484–502, pp. 28–29). Finally, the potential use of the state-of-the-art methods has been discussed in the Discussion (lines 503–521, pp. 29–30), in comparison with the proposed method.

Regarding the issues of low contrast in phase contrast micrographs, complex and different structures varying in time, and touch or overlap we have addressed in the revised Introduction and Discussion:

- Low contrast in phase contrast micrographs: This issue has been addressed with citations reporting image analysis methods capable of detecting the low contrast associated with small phase shifts [29 (Tsai et al, Software X, 2019), 30 (Al-Zaben et al., Sci. Rep., 2019)] (lines 81–84, p. 5). Accurately capturing low contrast thin lamellipodia while remaining insensitive to subcellular structures such as the organelles is a requirement for the identification of fine-grained characteristics in the mesenchymal migration modes, as discussed in the Discussion (lines 496–502, p. 29).

- Complex and different structures varying in time: Detection of “complete cell tracks at 1-min intervals for over 1 h, corresponding period from one peak to the next in two-speed mode migration of cancerous HT1080 fibrosarcoma (lines 490–491, p. 29)” is a requirement for distinguishing between the mesenchymal migration modes of cancerous fibrosarcoma and non-cancerous fibroblast cell types on flat tissue culture polystyrene. The significance of avoiding fragmentation errors has been addressed along with the importance of the label-free image acquisition method in the Discussion (lines 491–495, p. 29).

- Touch or overlap: Cell–cell contact is an important issue that supports the basis of characterization and discrimination of heterogeneous mesenchymal mode migrations rather than just for a segmentation method. We feel a deep interest in the influence of cell–cell contact, but it is beyond the scope of this study, which focuses on the discrimination of two types of mesenchymal mode migration with motile parameters extrac

---

## [Decision Letter · Decision Letter 1]

18 Feb 2025

Development of label-free cell tracking for discrimination of the heterogeneous mesenchymal migration

PONE-D-24-28597R1

Dear Dr. Miyoshi,

We’re pleased to inform you that your manuscript has been judged scientifically suitable for publication and will be formally accepted for publication once it meets all outstanding technical requirements.

Kind regards,

Ruo Wang

Academic Editor

PLOS ONE

Additional Editor Comments (optional):

Reviewers' comments:

Reviewer's Responses to Questions

**Comments to the Author**

1. If the authors have adequately addressed your comments raised in a previous round of review and you feel that this manuscript is now acceptable for publication, you may indicate that here to bypass the “Comments to the Author” section, enter your conflict of interest statement in the “Confidential to Editor” section, and submit your "Accept" recommendation.

Reviewer #1: All comments have been addressed

Reviewer #3: All comments have been addressed

2. Is the manuscript technically sound, and do the data support the conclusions?

Reviewer #1: Yes

Reviewer #3: Yes

3. Has the statistical analysis been performed appropriately and rigorously? 

Reviewer #1: Yes

Reviewer #3: Yes

4. Have the authors made all data underlying the findings in their manuscript fully available?

Reviewer #1: Yes

Reviewer #3: Yes

5. Is the manuscript presented in an intelligible fashion and written in standard English?

Reviewer #1: Yes

Reviewer #3: Yes

6. Review Comments to the Author

Reviewer #1: (No Response)

Reviewer #3: Authors commented all reviewer comments. The manuscript imporved significantly and is now ready to be accpeted.

7. PLOS authors have the option to publish the peer review history of their article (what does this mean? ). If published, this will include your full peer review and any attached files.

**Do you want your identity to be public for this peer review?** For information about this choice, including consent withdrawal, please see our Privacy Policy .

Reviewer #1: No

Reviewer #3: **Yes: ** David Dannhauser

---

## [Editor Report · Acceptance letter]

PONE-D-24-28597R1

PLOS ONE

Dear Dr. Miyoshi,

I'm pleased to inform you that your manuscript has been deemed suitable for publication in PLOS ONE. Congratulations! Your manuscript is now being handed over to our production team.

Kind regards,

on behalf of

Dr. Ruo Wang

Academic Editor

PLOS ONE